

# Genome-wide identification of the *Gossypium hirsutum* CAD gene family and functional study of *GhiCAD23* under drought stress

Xin Zhang[1], Ziyu Wang[1], Xingyue Zhong[1], Wanwan Fu[1], Yuanxin Li[1], Yunhao Liusui[1], Yanjun Guo[1], JingBo Zhang[1] and Bo Li[2]

[1] College of Life Science, Xinjiang Normal University, Urumqi, XinJiang, China
[2] Institute of Nuclear and Biological Technology, Xinjiang Academy of Agricultural Sciences, Urumqi, XinJiang, China

## ABSTRACT

Cinnamyl alcohol dehydrogenase (CAD) is a crucial enzyme in the final stage of lignin monomer biosynthesis. This study focuses on the CAD gene family within *Gossypium hirsutum*. Through comprehensive genomic analysis, we identified 29 *GhiCAD* genes within the *Gossypium hirsutum* genome using a bioinformatics approach. Phylogenetic analysis revealed that the GhiCAD family can be categorized into four subgroups, which are closest to the evolutionary relationship with *Arabidopsis thaliana*. There are multiple *cis*-acting elements on the promoters of *GhiCAD* genes associated with abiotic stress responses. Some *GhiCAD* genes demonstrated high expression in various tissues like root, leaf, and sepal, as well as in fiber and ovule at different developmental stages (10 days post anthesis (DPA), 15 DPA, 20 DPA, 25 DPA). The transcript levels of *GhiCAD23* were notably elevated when exposed to PEG treatment and drought stress (DS). *GhiCAD23* is also co-expressed with many known drought response genes, suggesting its involvement in the plant's reaction to DS. Employing virus-induced gene silencing (VIGS) technology to silence the *GhiCAD23* gene, it was found that silencing *GhiCAD23* reduced the tolerance of cotton to DS. Under DS, the relative leaf water content, superoxide dismutase (SOD), and catalase (CAT) enzyme activities of the *GhiCAD23*-silenced cotton plants were decreased by 31.84%, 30.22% and 14.19%, respectively, while malondialdehyde (MDA) was increased by 72.16% compared with the control cohort. Drought promotes the accumulation of lignin, and it was found that silencing the *GhiCAD23* reduces lignin accumulation in cotton under DS. The analysis of phenotypic and physiological indicators indicates that *GhiCAD23* is vital in cotton's resistance to DS. This investigation provides an important reference for future comprehensive exploration of the *GhiCAD23* gene's function in cotton's DS response mechanism.

Corresponding authors
JingBo Zhang,
18910445207@163.com
Bo Li, lbharrywei@sina.com

## INTRODUCTION

Cotton is an important cash crop that is typically grown in arid and semi-arid regions, where its cultivation relies entirely on irrigation. With the increasing scarcity of water for agriculture worldwide, cotton cultivation is highly vulnerable to drought stress (*Lin et al., 2024*). Drought stress (DS) can severely affect cotton growth, leading to reduced yield and quality (*Gao et al., 2021*). Throughout their long evolutionary history, plants have developed a complex metabolic regulatory network that adjusts the accumulation of certain metabolites to adapt to various environmental changes (*Sharma et al., 2024*). Lignin is an important secondary metabolite in plants. As a crucial structural element in the secondary cell walls of vascular plants, it provides mechanical strength and hydrophobicity to plant tissues, playing a key role in plant structure and facilitating the transport of water and minerals throughout the plant (*Weng & Chapple, 2010*). Lignin primarily derives from the dehydrogenative polymerization of three lignin monomers, namely coniferyl alcohol yielding guaiacyl (G)-lignin, sinapyl alcohol producing syringyl (S)-lignin, and p-coumaryl alcohol generating p-hydroxyphenyl (H)-lignin (*Vanholme et al., 2010*; *Fraser & Chapple, 2011*). As a crucial constituent of the plant cell wall, lignin forms a unique complex with cellulose and hemicellulose, constituting the plant's framework (*Mellerowicz et al., 2001*; *Gosselink et al., 2004*). Simultaneously, lignin fills the spaces between cellulose microfibrils in the cell wall, imparting rigidity to the plant cell wall (*Zhang et al., 2021*).

Research has demonstrated that lignin plays a crucial role in plant defense against environmental stressors (*Tripathi et al., 2003*). Under DS conditions, both ABA and lignin content in maize are significantly upregulated to enhance drought tolerance to DS (*Jiao et al., 2024*). The NAC transcription factor OsNAC5 mediates rice's drought resilience by modulating lignin deposition (*Bang et al., 2022*). Overexpression of *PaLectinL7* increases the lignin content of sweet cherries and enhances salt stress tolerance (*Wu et al., 2023*). The above studies indicate that lignin is involved in the process of plant resistance to abiotic stress.

The biosynthesis of lignin can be categorized into three primary phases: the synthesis of monolignols *via* the phenylpropanoid pathway in the cytosol, the transport of lignin precursors to the apoplast, and their subsequent polymerization within the cell wall (*Vanholme et al., 2010*; *Fraser & Chapple, 2011*; *Xie et al., 2018*). The three stages of lignin synthesis in plants are mainly catalyzed by various enzymes, including phenylalanine ammonia lyase (PAL), cinnamate 4-hydroxylase (C4H), p-coumarate 3-hydroxylase (C3H), caffeic acid O-methyltransferase (COMT), 4-coumarate-CoA ligase (4CL), cinnamoyl-CoA reductase (CCR), cinnamyl alcohol dehydrogenase (CAD), peroxidase (PRX), and laccases (LACs) (*Vanholme et al., 2019*).

Recent studies have elucidated the functions of certain enzymes involved in lignin synthesis in plant drought responses. Overexpression of *FuPAL1* from *Fritillaria unibracteata* significantly increased drought tolerance in transgenic Arabidopsis (*Qin et al., 2022*). Similarly, overexpression of *RgC4H* in *Rehmannia glutinosa* enhanced its tolerance to DS (*Yang et al., 2021*). Elevated expression of *NtCOMT1* resulted in reduced

wilting in genetically modified tobacco plants under water deprivation (*Yao et al., 2022b*). Tobacco overexpressing *Fm4CL2* exhibited greater drought tolerance compared to the wild type under DS conditions (*Chen et al., 2020*). Genetically modified rice specimens with elevated expression of *OsCCR10* exhibited improved resilience to water scarcity throughout developmental phases, accompanied by increased photosynthetic capacity, reduced water depletion rates, and elevated lignin levels in root structures relative to non-modified controls (*Bang et al., 2022*). CAD, the key and final enzyme in monolignol biosynthesis, catalyzes the synthesis of G, H, and S lignin units in the presence of NADPH (*Jiang et al., 2022*). To date, many CAD gene families in plants have been identified and analyzed, and some functions of *CAD* genes have been documented (*Yusuf et al., 2022*; *Hu et al., 2022*). Overexpression of *GsCAD1* from wild soybean in cultivated soybean effectively enhances resistance to mosaic virus (*Xun et al., 2022*). Introduction of *CAD1* from *Physcomitrium patens* into *Arabidopsis* increases lignin content, resistance to pathogens, and results in thicker roots and early flowering (*Jiang et al., 2022*). However, most studies on *CAD* genes have focused on their functions in plant growth, development, and defense against biotic stress, with limited reports on their functions in abiotic stress. Therefore, this research seeks to investigate the function of *CAD* genes in plant drought responses to further enhance our understanding of the functional diversity of the plant CAD gene family.

In this study, we employed the cotton genome data to systematically identify and characterize the CAD gene family, focusing on phylogenetic relationships and promoter *cis*-element compositions. We analyzed tissue-specific expression patterns of CAD gene family members using transcriptomic data. Integrating transcriptomic data under PEG treatment, qPCR, and co-expression analysis, we screened CAD gene family members involved in drought stress responses in cotton. Subsequently, we employed VIGS technology to silence selected candidate genes, allowing us to assess phenotypic and physiological variations between control and gene-silenced plants under drought stress conditions. This investigation is the first to elucidate the functional role of *CAD* genes in cotton's response to drought stress and provides valuable genetic resources for the molecular breeding of drought-resistant cotton varieties.

## MATERIALS AND METHODS

### Plant materials and growth conditions

The cotton seeds (*Gossypium hirsutum* cv. Zhongmian 113) were sterilized in 75% (v/v) ethanol for 5 min, then rinsed three times with distilled water. They were subsequently immersed in a 1% (v/v) NaClO solution for 10 min, rinsed again with distilled water, placed between two damp filter papers, and germinated in darkness at 28 °C for 2 days. The seedlings were then planted in pots containing a mixture of nutrient soil and vermiculite (3:1 by volume). The soil in each pot was pre-weighed to ensure a consistent soil volume. Following this, the pots were relocated to a greenhouse maintained at 25 °C/22 °C (day/night) with a 16-h light/8-h dark cycle.

## RNA extraction and quantitative analysis of gene expression

Treatment of 3-week-old seedlings of upland cotton variety Zhongmian 113 was carried out, dividing them into two cohorts: a control cohort with usual watering and a drought-treated cohort. The relative soil water content (RSWC) was used to assess the level of drought stress experienced by the cotton plants. Prior to the onset of drought, each pot is allowed to fully absorb the water, and the total weight of the pots is measured. The saturated weight of the soil is calculated by subtracting the weight of the empty pots from the total weight of the pots. The calculation formula for the RSWC (%) = (measured soil wet weight − dry weight)/(saturated weight − dry weight) × 100%. When the relative soil moisture content of the drought-treated cohort dropped below 50%, cotton leaves from both the normal watering cohort and the drought-treated cohort were collected. RNA was isolated from the cotton leaf samples using a plant RNA extraction kit (FOREGENE, Chengdu, China), followed by reverse transcription of the RNA into cDNA using a reverse transcription kit (FOREGENE, Chengdu, China). The UYS900 system was used for the qPCR experiments, with the specific reaction program as follows: 95 °C for 3 min, followed by 40 cycles of 95 °C for 10 s, 58 °C for 10 s, 72 °C for 20 s, 95 °C for 15 s, 58 °C for 1 min, and 95 °C for 15 s. qPCR was performed utilizing specific primers for *CAD* genes (Table S1) with a double-strand chimeric fluorescent dye (SYBR Green I), and the *GhHIS3* gene served as the reference gene. The qPCR primer sequence of *GhiCAD23* were obtained from this website (https://qprimerdb.biodb.org/organisms). The average of three biological replicates was taken, and gene expression levels were computed utilizing the $2^{-\Delta\Delta CT}$ approach.

## Identification and physicochemical property analysis of the cotton CAD gene family members

To locate the CAD gene family members within the upland cotton genome, Arabidopsis CAD protein sequences were obtained from TAIR (https://www.arabidopsis.org). The amino acid sequences of the AtCAD family (AT2G21890.1, AT2G21730.1, AT3G19450.1, AT4G34230.1, AT4G37970.1, AT4G37980.1, AT4G37990.1, AT4G39330.1, AT1G72680.1) were used as probes to search the upland cotton genome using the BLASTp program with an E-value threshold of $1e^{-5}$. Concurrently, the HMM model files of the CAD family (PF08240, PF00107) were retrieved from the Pfam protein domain database (http://pfam.xfam.org), and the upland cotton CAD family members were identified using HMMER with parameters set to E-value = $1e^{-5}$. The protein sequences of the potential CAD family members identified from BLAST and HMMER analyses were submitted to Cotton MD (https://yanglab.hzau.edu.cn/CottonMD/gene_search.1) for annotation, resulting in the identification of upland cotton CAD gene family members, designated as *GhiCAD1-GhiCAD29* based on their chromosomal positions. The essential physicochemical properties of the proteins were examined using the TBtools software. The GhiCAD protein sequences were submitted to the WOLF PSORT (https://wolfpsort.hgc.jp) database to predict the intracellular localization of GhiCAD family constituents.

## Systematic evolutionary analysis

The CAD protein sequences from *Gossypium hirsutum*, *Arabidopsis thaliana*, *Zea mays*, *Brachypodium distachyon*, and *Oryza sativa* were aligned using MEGA 11 for multiple sequence alignment. Subsequently, the alignment results were then used to construct a phylogenetic tree with FastTree. The resulting unrooted tree was uploaded to Evolview (http://evolgenius.info/evolview-v2/#mytrees/SHOWCASES/showcase%2001) to generate a circular phylogenetic tree.

## Analysis of promoter *cis*-acting elements

The 2,000 bp region preceding the initiation codon of the *GhiCAD* gene was obtained as the promoter sequence and analyzed for *cis*-regulatory elements using PlantCARE (http://bioinformatics.psb.ugent.be/webtools/plantcare/html/). Subsequently, TBtools software was utilized to visually represent the distribution of *cis*-acting elements within the CAD gene promoter.

## Expression analysis of CAD gene family members in *Gossypium hirsutum*

Transcriptome data for *GhiCAD* genes from 18 samples, encompassing various tissues, ovule developmental phases, and fiber growth stages in *Gossypium hirsutum*, were procured from the Cotton MD database (https://yanglab.hzau.edu.cn/CottonMD/heatmap.1). The expression levels of the transcriptome data were standardized to $\log_2$TPM values. TBtools software was used to generate expression heatmaps. Transcriptome sequencing was performed on cotton leaves treated with 18% PEG6000 hydroponics for 4 h, the data have been deposited in the OMIX, China National Center for Bioinformation/Beijing Institute of Genomics, Chinese Academy of Sciences (https://ngdc.cncb.ac.cn/omix: accession no. OMIX007375) (*CNCB-NGDC Members and Partners, 2024*). The expression data of CAD genes was extracted from the transcriptome, and then a heatmap was generated using TBtools.

## Gene co-expression network analysis

A total of 26 transcriptome datasets of *Gossypium hirsutum* under abiotic stress conditions were obtained from the Cotton MD database (https://yanglab.hzau.edu.cn/CottonMD/heatmap.1). The WGCNA package in R was used to build a co-expression network for these 26 transcriptome datasets, and the gene expression correlations were calculated. The genes that were co-expressed with the *GhiCAD23* gene were identified and functionally annotated. Cytoscape software was subsequently employed to visualize the network.

## Virus induced gene silencing (VIGS)

The VIGS experiments utilized the binary expression vectors TRV1 and TRV2 of the tobacco rattle virus (TRV). The cotton *CLA* gene, which is involved in chloroplast development and highly conserved throughout evolution, exhibits a distinct albino phenotype when silenced, thus serving as the positive control for VIGS experiments. A combination of Agrobacterium cultures harboring the TRV1 vector and the TRV2:*GhCLA*

recombinant vector was injected into cotton cotyledons to generate positive control plants designated as TRV2:*GhCLA*. A 400 bp sequence of the *GhiCAD23* gene was selected as the target sequence, amplified from a cotton cDNA library to construct the TRV2:*GhiCAD23* recombinant vector. Mixtures of Agrobacterium cultures containing TRV1 vector with TRV2 empty vector, TRV2:*GhCLA*, and TRV2:*GhiCAD23* recombinant vector were injected into cotton cotyledons. Plant lines injected with TRV1 and TRV2 served as control plants, denoted as TRV2:00, while plant lines injected with TRV1 and TRV2:*GhiCAD23* were considered silenced for the *GhCAD23* gene, labeled as TRV2:*GhiCAD23*. Appearance of albino phenotype in TRV2: *GhCLA* plants indicated successful silencing of the target gene. RNA was extracted from the leaves of TRV2:00 and TRV2:*GhiCAD23* plants for qPCR experiments to assess the effective silencing of the *GhiCAD23* gene, using *GhHIS3* as the reference gene. Expression levels of the target genes were calculated using the $2^{-\Delta\Delta Ct}$ method and averaged from three biological replicates (Table S1).

## Physiological parameters analysis of cotton plants under DS

Before initiating drought treatment, cotton plants with consistent growth were selected from TRV2:00 and TRV2:*GhiCAD23* lines and placed under identical conditions, receiving water every 3 days. On the 4 days after the final watering, drought treatment began, marking this day as day one of the drought treatment. After 18 days of period of natural drought, phenotypic observations and photographs of the TRV2:00 and TRV2:*GhiCAD23* cotton plants were taken. Leaves from both plant lines were collected for analysis of superoxide dismutase (SOD), catalase (CAT), and malondialdehyde (MDA) enzyme activities using visible spectrophotometry. The specific procedures for these measurements were in accordance with the instructions of the biochemical index assay kit (Grace, Suzhou, China; http://www.geruisi-bio.com/pro/index/classid/1). Samples were prepared according to the manufacturer's instructions. Total CAT activity was assayed by measuring the rate of decomposition of $H_2O_2$ at 510 nm. MDA contents were determined using thiobarbituric acid aspreviously described (*Wang et al., 2017*). SOD activity was determined by measuring the percentage of inhibition of the pyrogallol autoxidation. Each physiological indicator was measured with three biological replicates, and the data from these three biological replicates were subjected to a one-way ANOVA in GraphPad Prism 8 to calculate the standard deviation of the three biological replicates. The T-test in GraphPad Prism 8 software was used to analyze whether there is a significant difference in physiological indicator data between TRV2:00 and TRV2:*GhiCAD23*. At the same time, GraphPad Prism 8 software was used to draw bar graphs.

## Measurement of lignin content

The lignin content was measured using a lignin content measurement kit according to the manufacturer's instructions (Solarbio, Beijing, China). In brief, cotton stems were dried to a constant weight at 80 °C, then were filtered through a 40-mesh sieve, and approximately 3 mg was weighed into a 1.5 mL vial. An acetyl bromide and acetic acid mixture was then added and incubated at 80 °C for 40 min. Post-centrifugation, the supernatant was

collected, 200 μL was introduced to a 96-well UV plate, and light absorption was evaluated at 280 nm to determine the lignin content.

## RESULTS

### Identification and systematic evolutionary analysis of CAD gene family members in *Gossypium hirsutum*

A total of 29 *CAD* genes were identified within the *Gossypium hirsutum* genome and designated based on their chromosomal positions: *GhiCAD1-GhiCAD29*. As shown in Table S2, the proteins encoded by *GhiCAD* genes comprise 118–378 amino acids, with molecular weights spanning from 12.7 to 44.0 kD and isoelectric points between 5.32 and 8.48. Subcellular localization prediction indicated that the majority of *GhiCAD* genes are located in the cytoplasm, while a few are found in peroxisomes, mitochondria, and the nucleus (Table S2).

To elucidate the phylogenetic associations within the GhiCAD gene family, a multiple sequence alignment was performed using the protein sequences of CAD genes from *Arabidopsis thaliana*, *Oryza sativa*, *Zea mays*, and *Brachypodium distachyon* with the CAD proteins from *Gossypium hirsutum*, and a systematic evolutionary tree was constructed. The evolutionary analysis showed that the 55 plant CAD family members were divided into four subfamilies, and the *Gossypium hirsutum* CAD family members were unevenly distributed among the four subfamilies. Group 1 included seven *GhiCAD* genes, Group 2 included four genes, and Group 4 included two genes, and Group 3 included the largest number of *GhiCAD* genes, totaling 16, indicating that Group 3 has expanded during the evolution of *Gossypium hirsutum*. It was also found that only *Gossypium hirsutum* CAD family members and *Arabidopsis thaliana* CAD family members were present in Group 2 and 3, suggesting that the *Gossypium hirsutum* CAD gene family is more closely linked to the *Arabidopsis thaliana* family (Fig. 1).

### Analysis of *cis*-acting elements in the *GhiCAD* gene promoters

To further elucidate the biological processes involved in *GhiCAD* genes, we examined *cis*-acting elements within the promoter sequences of the 29 *GhiCAD* genes. Our findings revealed that the promoters of *GhiCAD* genes contain numerous *cis*-acting elements associated with environmental stress responses, including the abscisic acid response element ARE, ABRE3a, and ABRE4; the gibberellin response element TATC-box; the methyl jasmonate response element CGTCA-box; the drought stress response elements DRE core, DRE1, and MBS; and the flavonoid biosynthesis element MBSI, indicating that the *GhiCAD* genes are related to environmental stress responses in cotton. *Cis*-acting elements linked to transcription factor regulation, like MYB, MYC, are also abundantly present in the promoters of *GhiCAD* genes, indicating that the transcription of *GhiCAD* genes may be influenced by multiple transcription factors (Fig. 2, Table S3).

### Tissue expression patterns of *CAD* genes in *Gossypium hirsutum*

To investigate the potential involvement of *Gossypium hirsutum CAD* genes in cotton growth and development, we analyzed published transcriptome data to examine the

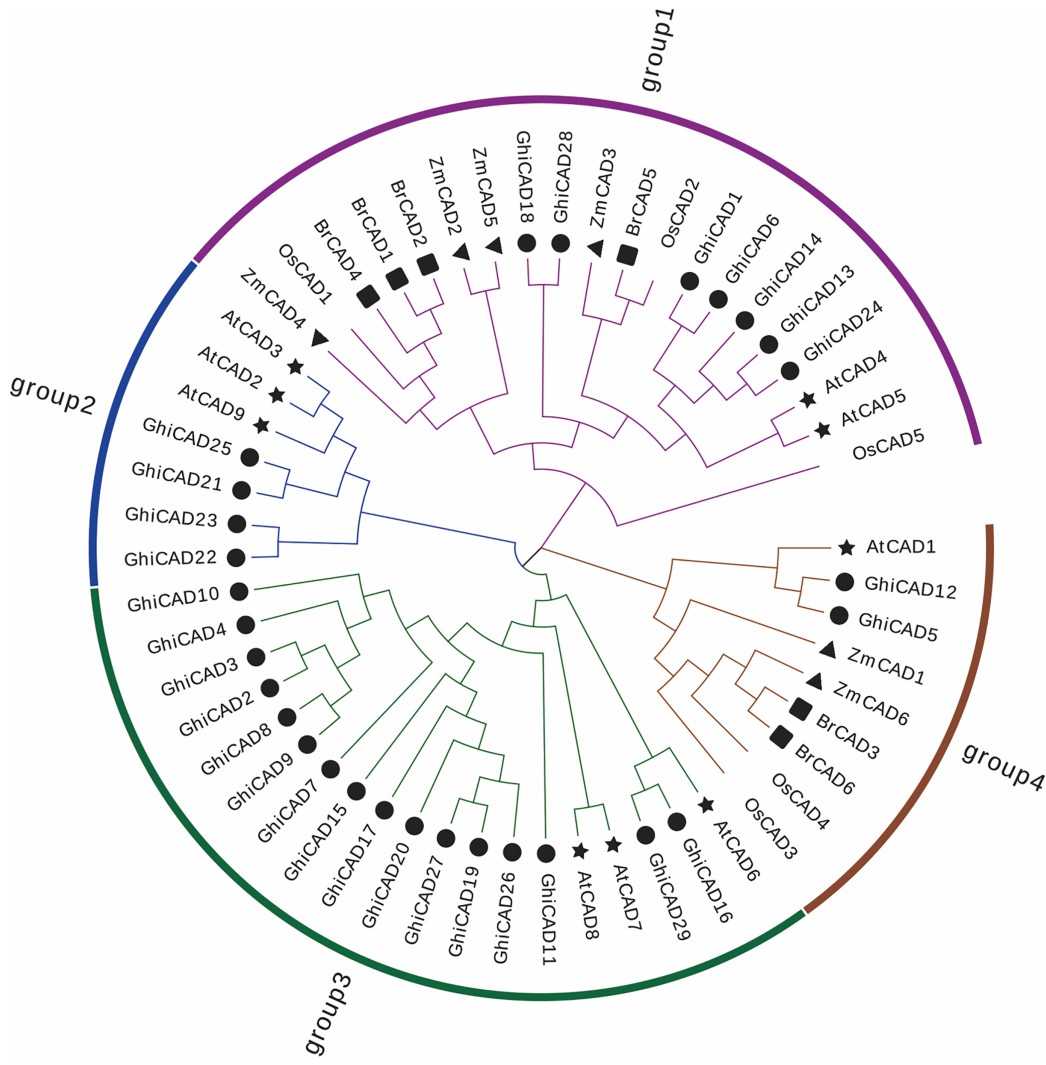

**Figure 1 The phylogenetic relationship of CAD proteins in *Gossypium hirsutum*, *Arabidopsis thaliana*, *Oryza sativa*, *Zea mays*, and *Brachypodium distachyon*.** The unrooted systematic evolutionary tree was constructed using FastTree by the maximum likelihood method with 1,000 bootstrap iterations. These four clusters depicted in distinct hues, while varied symbols denote genes originating from diverse plant taxa.

expression of 29 *GhiCAD* genes across various tissues, including roots, leaves, sepals, stems, anthers, bracts, filaments, pistils, ovules, and fibers at different developmental stages. The growth phases of cotton ovules and fibers are measured in days post anthesis (DPA). As shown in Fig. 3 and Table S4, certain *CAD* gene families, such as *GhiCAD10*, *GhiCAD25*, and *GhiCAD21*, are highly expressed in multiple tissues, including roots, leaves, sepals, ovules, and fibers at various developmental stages. This indicates that *GhiCAD10*, *GhiCAD25*, and *GhiCAD21* play roles in the development of several cotton tissues. Conversely, some *CAD* genes exhibit tissue-specific expression. For instance, *GhiCAD23* is predominantly expressed in roots, *GhiCAD4* in pedicels, and *GhiCAD18* in filaments and stamens. These observations imply that these latter genes primarily contribute to the developmental processes of specific tissues.

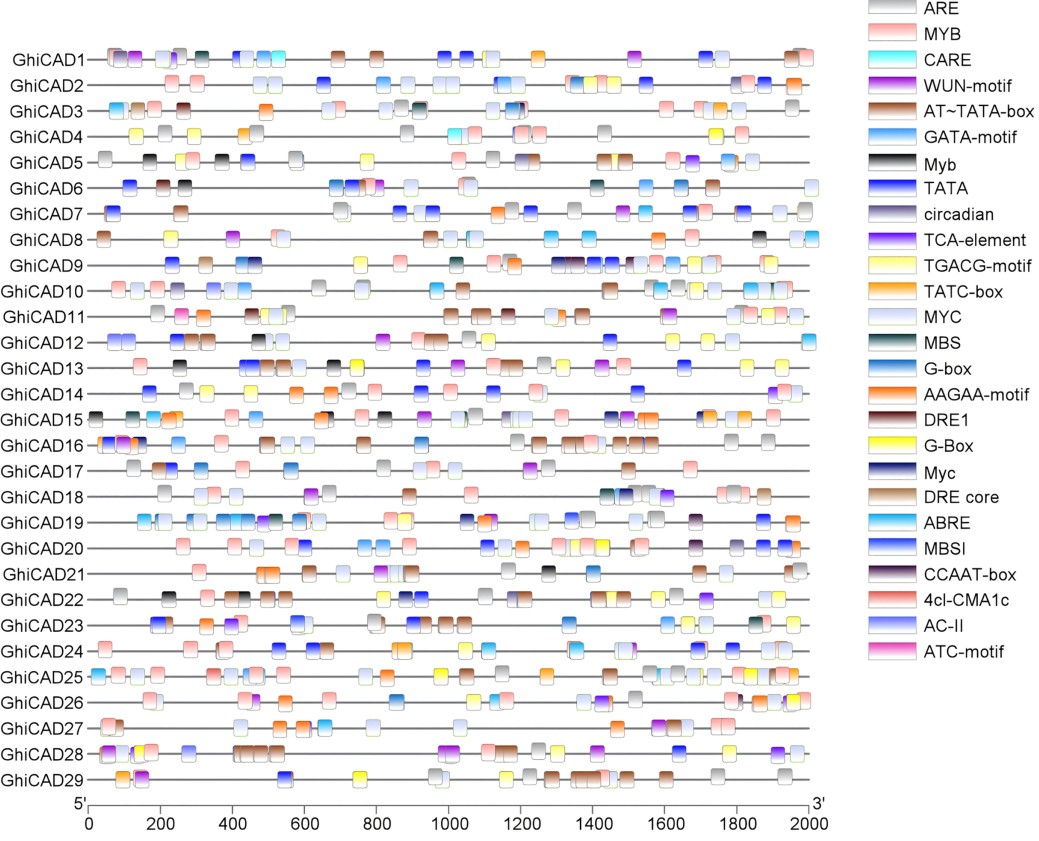

**Figure 2 Analysis of *cis*-acting elements on CAD gene promoters in *Gossypium hirsutum*.** Different small boxes in the figure represent different promoter *cis*-acting elements, and the small boxes labelled at the top right represent *cis*-acting elements.

## Expression analysis of *CAD* genes in *Gossypium hirsutum* under DS

To further investigate the potential function of *CAD* genes in cotton's response to DS, we examined the expression profiles of CAD gene family members under conditions of PEG treatment using transcriptome data. The data indicated that most *GhiCAD* genes did not show significant changes in expression levels before and after PEG treatment. However, the expression of *GhiCAD10* and *GhiCAD23* changed significantly under PEG treatment conditions. *GhiCAD23* was markedly elevated by PEG, suggesting that the *GhiCAD23* gene might implicated in the cotton's response to DS (Fig. 4, Table S5).

To further examine whether *GhiCAD23* responds to DS, we utilized qPCR to evaluate the expression of *GhiCAD23* in cotton plants under normal watering and drought treatment conditions. The results showed that the transcription level of *GhiCAD23* was significantly higher in drought-treated cotton plants compared to normally watered plants, indicating that *GhiCAD23* is implicated in cotton's resistance to DS (Fig. S1).

## Co-expression network analysis of the *GhiCAD23* gene

We conducted a co-expression network analysis of the *GhiCAD23* gene and performed functional annotation of the genes co-expressed with *GhiCAD23*. The analysis indicated

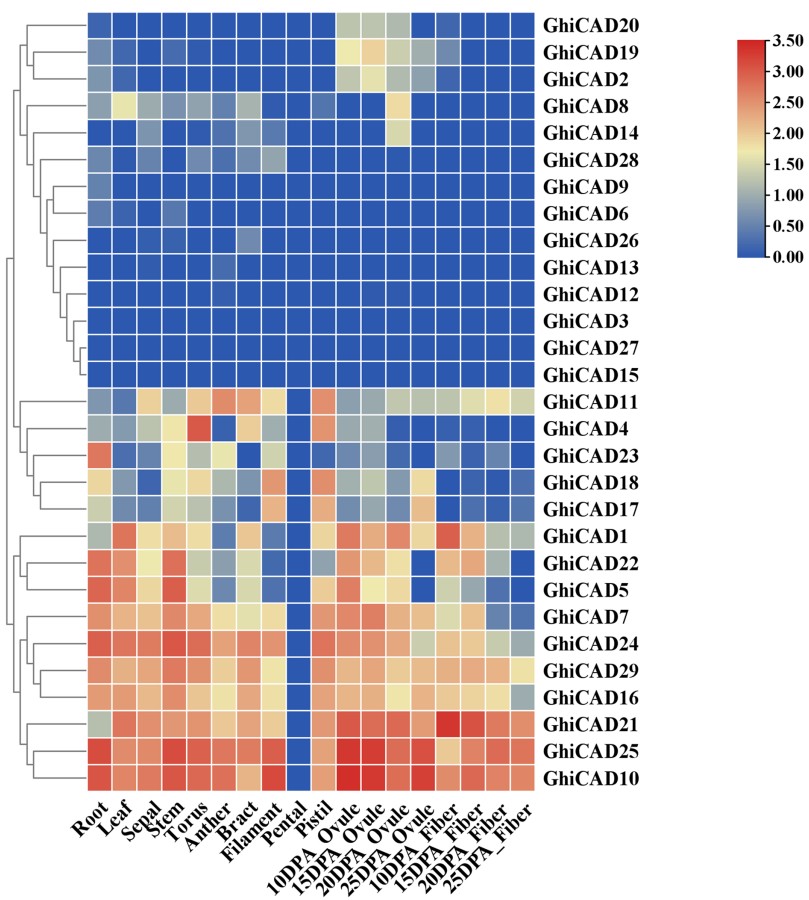

**Figure 3** **Tissue expression analysis of *CAD* genes in *Gossypium hirsutum*.** Each small square displays the $\log_2$ TPM values of *CAD* genes in different samples, with red or blue indicating the difference in expression levels in each sample.

that *GhiCAD23* displays expression patterns similar to several known drought-responsive genes, further corroborating that *GhiCAD23* is related to DS response in cotton (Fig. 5).

## Knocking down the *GhiCAD23* gene decreases the drought tolerance of cotton

The gene was silenced using VIGS methodology to further examine the function of the *GhiCAD23* gene in DS response in cotton. When positive plants (TRV2: *GhCLA*) exhibited albino symptoms, it indicated effective silencing of the target gene (Fig. 6A). Quantitative polymerase chain reaction (qPCR) was utilized to evaluate the expression level of *GhiCAD23* in TRV2:00 and TRV2:*GhiCAD23* plants, confirming the effective silencing of the *GhiCAD23* gene. The findings revealed a significantly lower transcription level of *GhiCAD23* in TRV2:*GhiCAD23* plants was markedly lower compared to that in TRV2:00 plants ($p < 0.01$), indicating effective suppression of *GhiCAD23* transcription (Fig. 6B). Subsequently, 3-week-old TRV2:00 and TRV2:*GhiCAD23* plants were subjected to drought treatment. At 0 d of drought, no notable phenotypic differences were discovered between TRV2:00 and TRV2:*GhiCAD23* plants. After 18 d of period of natural drought,

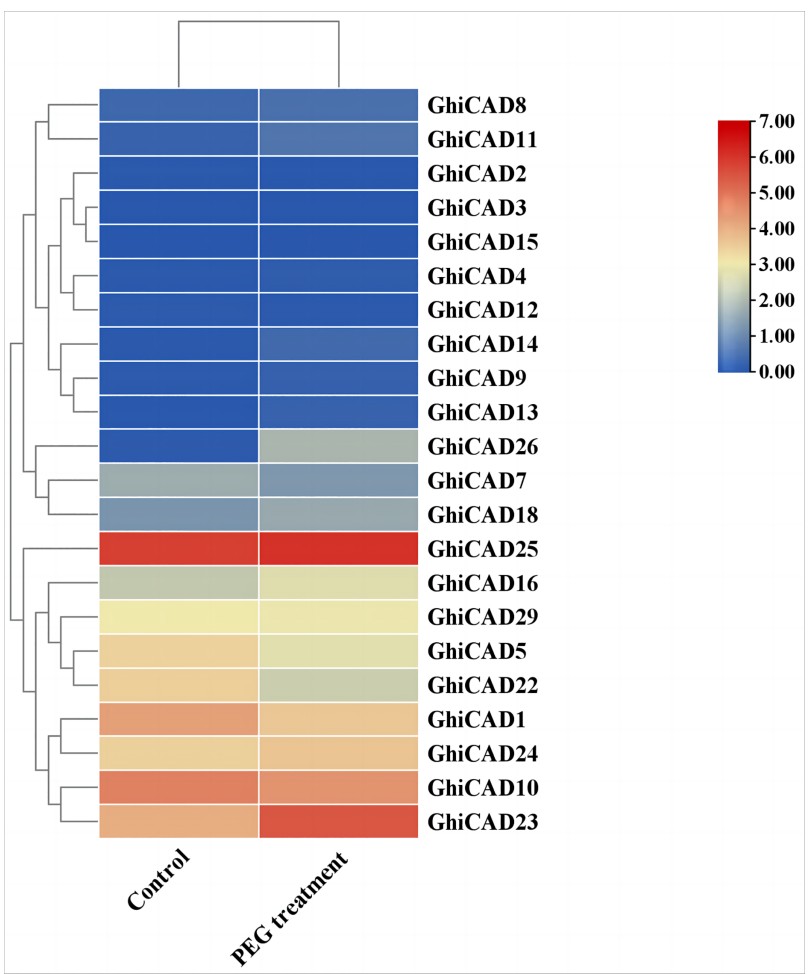

**Figure 4 Expression pattern of *GhiCAD* genes under PEG treatment.** Each small square displays the log2 FPKM values of *CAD* genes in the control group and the PEG treatment group, with red or blue indicating the difference in expression levels in each sample.

TRV2:00 plants displayed noticeably superior development compared to TRV2:*GhiCAD23* plants.

Additionally, the wilting rate was observed to be 33.33% in TRV2:00 plants, while TRV2:*GhiCAD23* plants exhibited a wilting rate of 100% (Fig. 6C). TRV2:00 not only had a lower wilt rate, but also a lower degree of wilt than TRV2:*GhiCAD23*.

Physiological parameters of TRV2:00 and TRV2:*GhiCAD23* plants were further assessed under DS conditions. The results revealed that the relative water content of TRV2:00 plant leaves was 31.84% higher relative to the TRV2:*GhiCAD23* plants, while the MDA content in TRV2:00 cotton plants was 72.16% lower than in TRV2: *GhiCAD23* plants (Figs. 6D, 6E). Additionally, the SOD and CAT contents in TRV2:*GhiCAD23* plants were reduced by 30.22% and 14.19%, respectively, compared to TRV2:00 plants (Figs. 6F, 6G). Both phenotypic observations and physiological parameter measurements suggest that silencing the *GhiCAD23* gene diminishes the drought resilience of cotton.

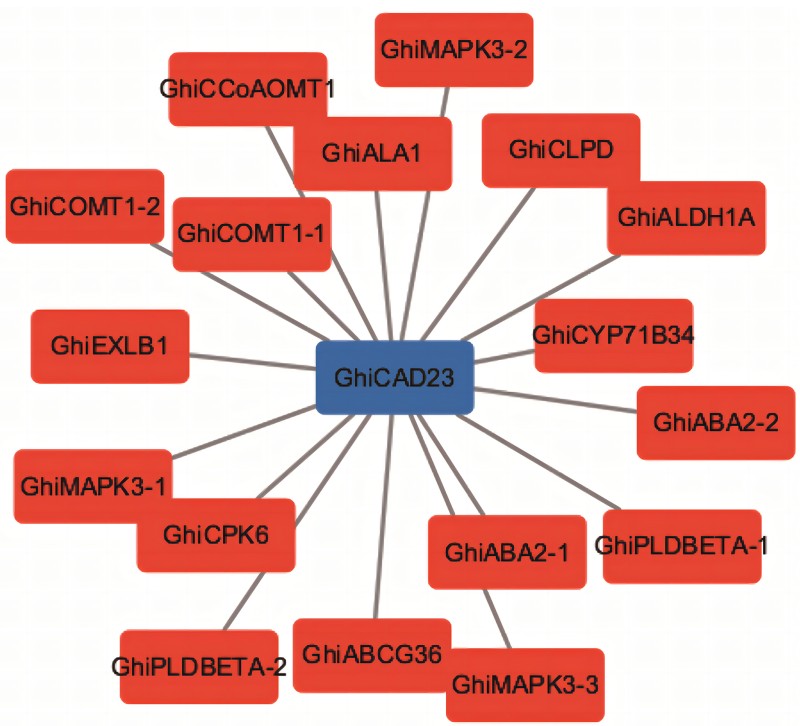

**Figure 5 Co-expression analysis of *GhiCAD23* gene.** Each red box represents a known drought-responsive gene.                                    

### *GhiCAD23* responds to DS by affecting the accumulation of lignin

To investigate determine whether the lignin content of cotton during the seedling stage changes under DS, we measured the lignin content in cotton plants under normal watering and DS conditions. Our results showed a significant increase in lignin content in cotton plants under DS compared to those under normal watering, indicating that DS significantly promotes lignin accumulation in cotton (Fig. 7A). Additionally, to assess the impact of silencing the *GhiCAD23* gene on lignin accumulation under DS, we measured the lignin content in TRV2:00 and TRV2:*GhiCAD23* plants under DS. The outcomes demonstrated that the lignin content in TRV2:*GhiCAD23* plants was reduced compared to the TRV2:00 plants, suggesting that silencing *GhiCAD23* reduces lignin accumulation in cotton plants under DS (Fig. 7B).

### DISCUSSION

CAD assumes a crucial role in lignin synthesis, responsible for catalyzing the final step in the lignin monomer biosynthesis pathway (*Weng & Chapple, 2010*; *Bagniewska-Zadworna et al., 2014*). Previous studies have systematically analyzed the CAD gene family in species like *Arabidopsis* (*Sibout et al., 2003*), *Populus* (*Barakat et al., 2009*), *Rice* (*Tobias & Chow, 2005*), and *Brachypodium distachyon* (*Bukh, Nord-Larsen & Rasmussen, 2012*). However, research on the cotton CAD gene family remains limited. In this study, we identified 29 *CAD* genes in cotton using bioinformatics methods and systematically analyzed the *GhiCAD* gene family, further expanding our understanding of the plant CAD gene family.

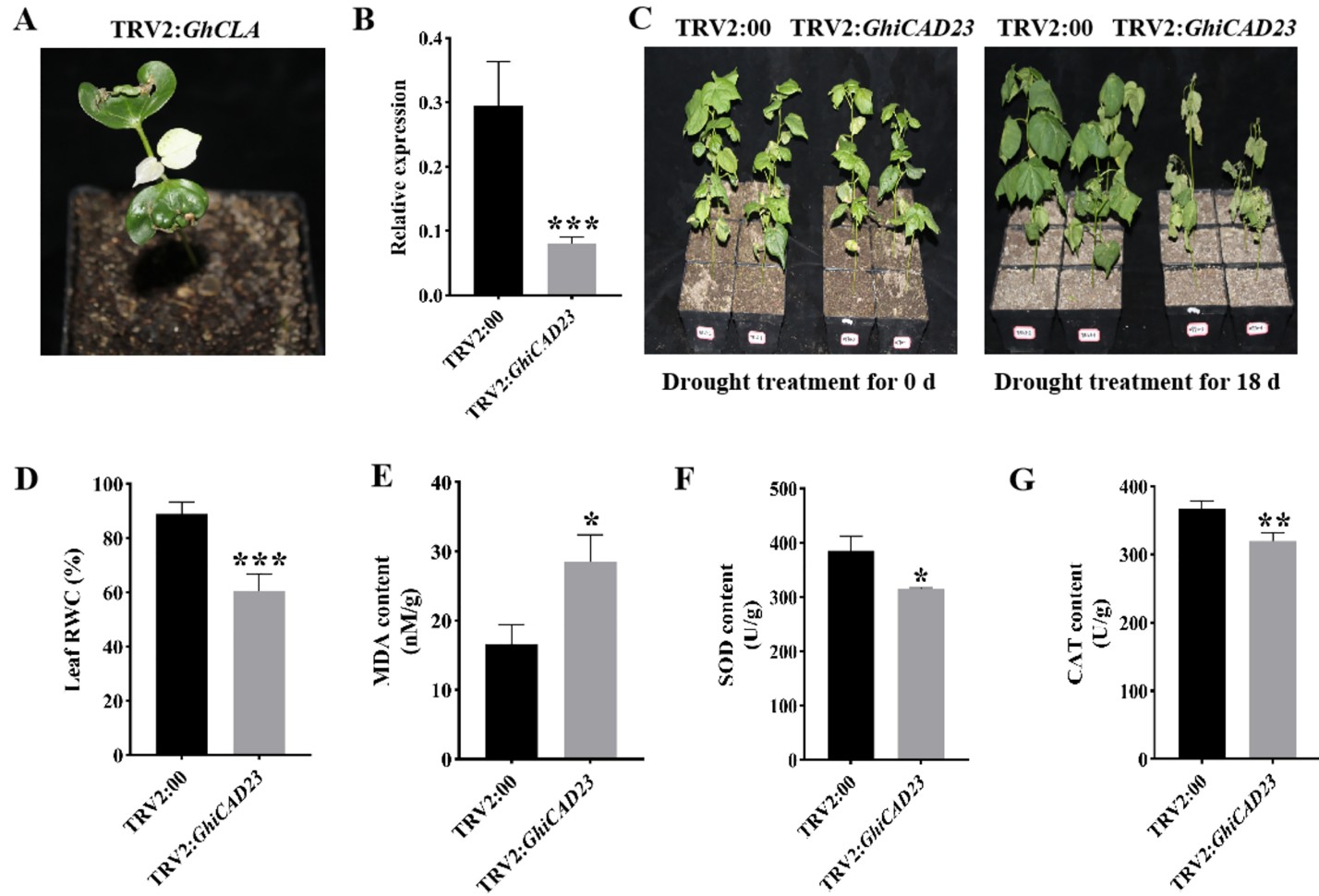

**Figure 6 Silencing of *GhiCAD23* reduced cotton tolerance to drought stress.** (A) Positive control plants. (B) Expression of *GhiCAD23* in TRV2:00 and TRV2:*GhiCAD23* plants. (C) Phenotype of TRV2:00 and TRV2:*GhiCAD23* plants under DS. (D) Relative water content (RWC) of TRV2:00 and TRV2:*GhiCAD23* plants under DS. (E–G) The content of MDA (E), SOD (F) and CAT (G) of TRV2:00 and TRV2:*GhiCAD23* plants under DS. SD represents the standard deviation derived from three separate trials. *$p < 0.05$; **$p < 0.01$; ***$p < 0.001$. Student's *t*-test.

Lignin plays a pivotal function in plant growth and development (*Wang et al., 2022*). It enhances the hydrophobicity of plant cell walls, promotes the development of plant tissues and organs, and is significant for plant adaptation to the environment (*Vanholme et al., 2019*). Lignin biosynthesis is notably enhanced under drought stress (*Xie, Cao & Xu, 2024*). For example, when *Glycine max* is subjected to DS, the expression level of the *CCoAOMT* gene in the elongation zone of soybean roots is significantly upregulated, leading to an increase in lignin content and thereby reducing water loss in soybean roots (*Yamaguchi et al., 2010*). *OsOLP1* augments drought resilience in rice by promoting lignin accumulation (*Yan et al., 2023*). Down regulation of microRNA 397 expression increases lignin accumulation in chickpea roots, thereby enhancing chickpea's tolerance to DS (*Sharma et al., 2023*). In maize, elevated expression of the peroxidase gene *ZmPRX1* promotes lignin accumulation in maize roots and increases seedling drought tolerance (*Zhai et al., 2024*). Our study found that DS significantly increases lignin content in cotton,

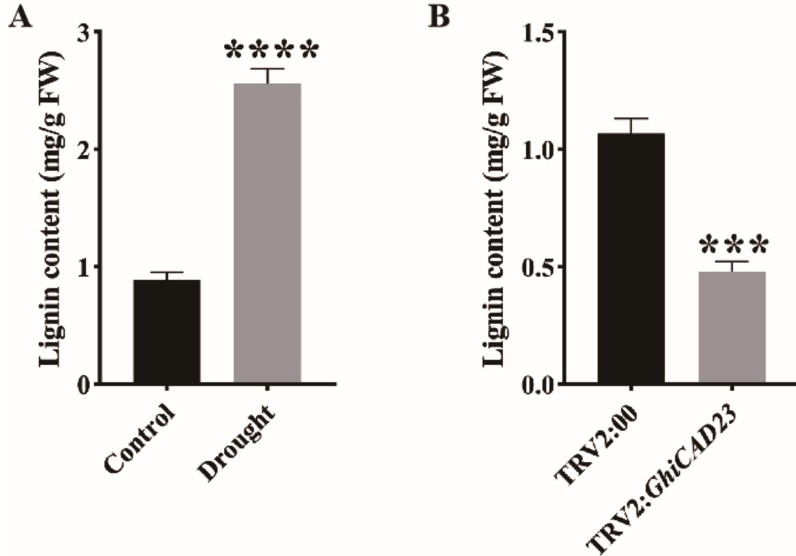

**Figure 7 Lignin content analysis.** (A) Lignin content in the stem of cotton plants was evaluated under normal watering and DS conditions. (B) Lignin contents in three-leaf stage TRV2:00 and TRV2:*Ghi-CAD23* cotton stems. SD represents the standard deviation derived from three separate trials. ***$p < 0.001$; ****$p < 0.0001$. Student's *t*-test.

further illustrating the close relationship between lignin accumulation and plant drought tolerance.

As a crucial rate-limiting enzyme in lignin biosynthesis, CAD is vital in plant defense against environmental stresses (*Kim et al., 2004*, *2007*; *Dong & Lin, 2021*). The transcription levels of multiple *NtCAD* genes in tobacco change under drought stress conditions (*Wu et al., 2024*). DS induces the expression of three melon *CAD* genes, *CmCAD1*, *CmCAD2*, and *CmCAD3*, to enhance lignin biosynthesis (*Jin et al., 2014*; *Liu et al., 2020*). Examination of *Phyllostachys edulis PheCAD* gene expression patterns under abiotic stress conditions showed that *PheCAD2* was notably elevated following abiotic stress exposure (*Vasupalli et al., 2021*). Our experimental data reveal that drought stress induces the accumulation of transcripts from multiple *GhiCAD* genes. Notably, *GhiCAD23* shows an expression pattern similar to several well-known drought response genes, suggesting that *CAD* genes play a role in the plant's response to drought stress.

*CAD-C* and *CAD-D* are the principal genes implicated in lignin synthesis in *Arabidopsis*, and they play a significant role in defense against the bacterial pathogen *Pseudomonas syringae pv.* tomato (*Tronchet et al., 2010*; *Sibout et al., 2005*). Transcription factors RhbZIP17 and RhWRKY30 enhance rose resistance to the pathogen *Botrytis cinerea* by activating the transcription of *RhCAD1* (*Li et al., 2024*). Previous studies have elucidated the function of *CAD* genes in plant biotic stress responses, while research on the role of *CAD* genes in plant abiotic stress responses has primarily focused on expression pattern analyses. Our study revealed the role of the *GhiCAD23* gene in cotton drought response, finding that suppression of this gene's expression diminished cotton's resilience to DS. This offers compelling support for the involvement of *CAD* genes in plant reactions

to environmental stressors and enhances our comprehension of plant *CAD* gene functionality.

Drought stress leads to the accumulation of excessive reactive oxygen species (ROS) within plants, which can cause oxidative damage to biomolecules and even cell death (*Li et al., 2023*). Plants use an enzymatic reaction system to remove excess ROS, with CAT and SOD being key enzymes in the ROS detoxification system (*Raza et al., 2021*; *Luo et al., 2024*). Under DS conditions, both SOD and CAT enzyme activities are lower in *GhiCAD23* gene-silenced plants compared to the control plants. This indicates that suppressing *GhiCAD23* gene expression weakens the cotton plant's ROS scavenging ability, leading to increased sensitivity to drought stress. Four-coumarate-CoA ligases (4CL) are crucial enzymes involved in lignin biosynthesis in plants. Silencing the cotton *Gh4CL7* gene weakens the plant's drought resistance, with lignin content in *Gh4CL7*-silenced plants decreasing by about 20% compared to control plants (*Sun et al., 2020*). Our research found that silencing the *GhiCAD23* gene reduces lignin accumulation in cotton plants under drought stress. This further indicates that lignin biosynthesis-related genes participate in the plant's response to drought stress by affecting lignin accumulation under such conditions. Many genes that encode catalytic enzymes also participate in the biological processes of plant growth, development, and stress resistance by influencing the expression of other genes (*Qin et al., 2017*; *Yao et al., 2022a*; *Zhang et al., 2024*; *Guo et al., 2024*). Therefore, we speculate that silencing the *GhiCAD23* gene may also weaken the drought resistance of cotton by affecting the expression of other genes.

Drought stress severely affects crop growth, yield, and quality. Exploring drought-resistant gene resources in crops and using genetic engineering techniques to enhance drought tolerance is an effective approach for developing drought-resistant germplasm resources (*Oladosu et al., 2019*). In this study, we identified a new drought-resistant gene in cotton, *GhiCAD23*, which serves as a potential target gene for the subsequent creation of new drought-resistant germplasm resources in cotton using genetic engineering techniques.

## CONCLUSIONS

This article identifies 29 *CAD* genes at the whole-genome level in cotton. Evolutionary analysis reveals that the CAD gene family can be divided into four subgroups. Multiple *cis*-acting elements related to plant abiotic stress responses are present in the promoters of CAD family genes. Members of the CAD gene family are expressed in various tissues of cotton, indicating their involvement in the developmental processes of these tissues. Through transcriptome data, qPCR, and co-expression analysis, we identified a *CAD* gene, *GhiCAD23*, that participates in cotton's response to drought stress. Further functional studies demonstrate that suppressing the *GhiCAD23* gene weakens cotton's drought resistance, reduces SOD and CAT enzyme activities under drought stress, and increases MDA accumulation. Additionally, silencing the *GhiCAD23* gene also decreases lignin accumulation in cotton under DS conditions. These results provide a basis for further understanding the role of CAD in plant resistance to drought stress. At the same time, this

study will contribute to future genomics-assisted breeding programs in cotton to improve its drought tolerance.

### Funding

This work was funded by the Natural Science Foundation of Xinjiang Uygur Autonomous Region (No. 2022D01B39) and the Youth Doctoral Project of the Talent Development Fund "Tianchi Youth Talent" Introduction Program in Xinjiang Uygur Autonomous Region. The funders had no role in study design, data collection and analysis, decision to publish, or preparation of the manuscript.

### Grant Disclosures

The following grant information was disclosed by the authors:
Natural Science Foundation of Xinjiang Uygur Autonomous Region: 2022D01B39.
Youth Doctoral Project of the Talent Development Fund "Tianchi Youth Talent" Introduction Program in Xinjiang Uygur Autonomous Region.

### Competing Interests

The authors declare that they have no competing interests.

### Author Contributions

- Xin Zhang conceived and designed the experiments, performed the experiments, analyzed the data, prepared figures and/or tables, and approved the final draft.
- Ziyu Wang performed the experiments, analyzed the data, prepared figures and/or tables, and approved the final draft.
- Xingyue Zhong performed the experiments, prepared figures and/or tables, and approved the final draft.
- Wanwan Fu performed the experiments, prepared figures and/or tables, and approved the final draft.
- Yuanxin Li performed the experiments, prepared figures and/or tables, and approved the final draft.
- Yunhao Liusui performed the experiments, prepared figures and/or tables, and approved the final draft.
- Yanjun Guo performed the experiments, prepared figures and/or tables, and approved the final draft.
- JingBo Zhang conceived and designed the experiments, analyzed the data, authored or reviewed drafts of the article, and approved the final draft.
- Bo Li conceived and designed the experiments, analyzed the data, authored or reviewed drafts of the article, and approved the final draft.

### Data Availability

The expression data under PEG treatment conditions is available at OMIX, China National Center for Bioinformation/Beijing Institute of Genomics: OMIX007375.

The raw measurements are available in the Supplemental Files.

## Supplemental Information

Supplemental information for this article can be found online at http://dx.doi.org/10.7717/peerj.18439#supplemental-information.

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
