# Peer review of "Genome-wide identification of the *Gossypium hirsutum* CAD gene family and functional study of *GhiCAD23* under drought stress"

_PeerJ, doi:10.7717/peerj.18439_

## Round 0.1 · original submission · Minor Revisions

Dear Authors
The manuscript cannot be accepted for publication in its current form. It needs a minor revision before publication. The authors are invited to revise the paper considering all the suggestions made by the reviewers. Please note that the requested changes are required for publication.
With Thanks

Reviewer 1 ·

Basic reporting

Thank you for considering me for reviewing the manuscript “Genome-Wide Identification of the Gossypium hirsutum CAD Gene Family and Functional Study of GhiCAD23 Under Drought Stress”. The manuscript presents valuable research on CAD gene family in Gossypium hirsutum, focusing on identification of 29 CAD genes and the functional characterization of GhiCAD23 in response to drought stress, making it a valuable contribution to plant molecular biology.

Suggestion:
Conduct a thorough English language edit of the manuscript to correct grammatical errors and improve clarity. Also, consider shortening lengthy sentences to enhance readability. Summarize redundant sentences and focus on highlighting the important aspects of the study.

Include relevant keywords that are not presented in the title and abstract to improve searchability.

The introduction focused on lignin biosynthesis and the role of CAD genes in different paragraphs which should be summarized and improved. It contains sentences that are overly complex and could be rewritten for clarity. Streamline the introduction by reducing redundancy and simplifying sentence structure. Lines 56-64 present the impact of silicon treatment on lignin content in tomato, as well as melatonin in wheat, which is not mainly relevant to the studied aspect. Additionally, the paragraph on cotton importance and the detrimental effects of drought stress, particularly in semi-arid regions, should be introduced earlier in the introduction. The research gap, hypothesis, and objectives need to be clearly defined to provide a strong foundation for the study.

The methods are well-described and well-structured. The description of the applied treatments could be more detailed regarding the control and experimental conditions. Specify the reference gene used for the normalization of qPCR. The statistical methods used to analyze the data should be clarified, number of replicates, statistical tests employed, and significance thresholds.
Line 123: Scientific name should be in italics as well as throughout the manuscript

The results are well-written and presented. Simplify the text by focusing on key findings and their implications.

The discussion and conclusion sections are brief and require significant enhancement to provide a more thorough and insightful analysis. To strengthen discussion section, the authors should integrate a more comprehensive comparison with existing literature, particularly by expanding their analysis to include relevant studies. Currently, there is some repetition of results that could be streamlined to improve the flow and maintain reader engagement. It would benefit from a deeper exploration of the underlying mechanisms of GhiCAD23 in drought stress. Additionally, the broader implications of these findings for cotton breeding programs should be more explicitly stated. The authors should focus on interpreting the results within the context of existing research, emphasizing the practical applications of their findings in cotton breeding. Moreover, the discussion would be more robust if it highlighted how this study advances the current understanding of CAD genes in plant stress responses. Identifying areas where further research is needed will also enhance the discussion, providing a clearer pathway for future investigations.

References: Cross-check all in-text citations with the reference list to ensure consistency. Update the references to include recent research in the field. Please review and standardize the reference section according to style guidelines. The first letter of every word in journal names should be capitalized not just the first word.

Experimental design

The experimental design is appropriate

Validity of the findings

The findings of this study are generally valid and contribute meaningfully to the literature on drought stress in Gossypium hirsutum.

Reviewer 2 ·

Basic reporting

No comment

Experimental design

The study demonstrates an acceptable experimental design.

Validity of the findings

No comments

Additional comments

The study provides a valuable contribution to understanding the role of the CAD gene in cotton's drought stress response. By combining bioinformatics, gene expression analysis, and functional validation through VIGS, the researchers have successfully identified and characterized a specific CAD gene (GhiCAD23) as a key player in cotton's drought tolerance. The findings are significant as they elucidate a potential molecular mechanism underlying cotton's drought resistance, which could inform future breeding strategies for developing drought-tolerant cotton cultivars.
-Comments and Suggestions for Authors
- Abstract
-The abstract provides a clear and concise overview of the research
- Consider adding specific numerical data, such as the percentage increase or decrease in gene expression or physiological parameters, to enhance the impact of the findings.
-While the abstract mentions lignin accumulation, it could briefly elaborate on the connection between lignin and drought stress tolerance.
¬- Consider emphasizing the novelty of the study, such as being the first to investigate the role of GhiCAD23 in cotton drought stress response.
- Introduction
- While the introduction provides a good background, it could be strengthened by more explicitly stating the specific objectives of the study. For example, the introduction could highlight the gap in knowledge regarding the role of CAD genes in cotton's drought response.
- The introduction could be further improved by organizing the information more logically. For instance, the section on the functions of CAD genes could be placed earlier in the introduction to better set the stage for the study's focus.
- Materials & Methods
- Some sections could benefit from a brief explanation of why specific methods were chosen (e.g., qPCR for gene expression analysis, VIGS for gene silencing).
- The description of drought treatment could be elaborated on. Specifying the initial soil moisture content and the target moisture level reached during drought stress would provide a clearer picture of the experiment's conditions.
- It would be helpful to mention the source or justification for using unpublished data on transcriptome analysis under PEG treatment.
- Results
-The results section provides a comprehensive overview of the study's findings, supporting the hypothesis that GhiCAD23 plays a crucial role in cotton's drought response.
- While the results describe significant differences between groups, it would be beneficial to mention the specific statistical tests (e.g., t-test, ANOVA) and p-values to strengthen the conclusions.
-Providing quantitative data for phenotypic observations (e.g., plant height, biomass) could enhance the understanding of the impact of GhiCAD23 silencing on plant growth.
- While the study includes positive controls for VIGS, it would be helpful to mention the use of negative controls (e.g., empty vector control) for gene expression and phenotypic analyses.
- Discussion
-While the study provides evidence for the involvement of GhiCAD23 in drought stress, further exploration of the underlying mechanisms could be beneficial. For example, the discussion could speculate on how GhiCAD23 might influence lignin composition or structure to enhance drought tolerance.
-The discussion could briefly touch upon the potential implications of these findings for developing drought-resistant cotton cultivars.
- Conclusion
The conclusion could be strengthened by briefly discussing the potential implications of these findings for cotton breeding and agriculture. For example, the conclusion could mention the possibility of using GhiCAD23 as a target for genetic improvement of drought tolerance.

·

Basic reporting

The article titled " Genome-Wide identiûcation of the Gossypium hirsutum CAD gene family and functional study of GhiCAD23 under drought stress " explores the comprehensive identification and functional analysis of CAD (cinnamyl alcohol dehydrogenase) genes in cotton, focusing on GhiCAD23 under drought conditions. The study provides a detailed examination of the gene family's role in drought tolerance, integrating phylogenetic analysis, gene expression profiling, and physiological assessments. Through a series of experiments conducted under controlled greenhouse conditions, the authors identified 29 CAD genes in the cotton genome, with GhiCAD23 showing significant involvement in drought response. The findings highlight the potential of GhiCAD23 as a target for genetic improvement in cotton, offering insights into its regulatory mechanisms and contributions to drought resilience. This research advances our understanding of the molecular basis of drought tolerance in cotton, with implications for developing more resilient crop varieties. However, before this article can be considered for publication in PeerJ, substantial revisions are necessary to enhance the clarity, accuracy, and completeness of the manuscript. These improvements include correcting spelling mistakes, clearly defining developmental stages, and providing detailed criteria for transcription level assessments, including TPM values. The abstract should be expanded to include full forms of abbreviations, a brief methodology, key percentage values in physiological traits. Additionally, the introduction needs to address spacing issues and update references to the latest research. Abbreviations should be listed, and scientific nomenclature should be italicized consistently throughout the manuscript. The materials and methods section requires detailed descriptions of greenhouse conditions, water stress imposition, soil characteristics, and the statistical design of the experiments. In the results section, key values indicating percentage increases or decreases in physiological and biochemical parameters should be included, along with clear descriptions of phylogenetic analysis and gene localization. The discussion section needs to be thoroughly rewritten, focusing on the authors' findings in the context of current literature. The conclusion should be broadened to validate the findings, and the manuscript should include a list of abbreviations and self-explanatory figures and tables. Finally, the authors should consider future recommendations regarding the silencing of GhiCAD genes and ensure that all references and in-text citations are accurate and up to date. By addressing these points, the manuscript can be significantly strengthened, making it a more valuable contribution to the field of plant genetics and drought tolerance research.

Specific Comments:
Abstract:
Line No. 30: There is a spelling error in the term "as-sociated." Please correct it.
Line No. 32: Clearly specify the developmental stages discussed in the study.
Line No. 32: Demonstrate the criteria used for assessing transcription levels, and at least mention the TPM values.
Abbreviations: Ensure that the full forms of all abbreviations (e.g., PEG, SOD, POD) are provided upon their first mention in the abstract. Additionally, the abstract and conclusion should be self-explanatory.
Methodology: The abstract currently lacks a description of the methodology. Please include a brief overview of the materials and methods used.
Key Values: Incorporate key values in the abstract, such as percentage increases or decreases in physiological traits, to provide a quantitative perspective on the results.
Keywords: The abstract is missing keywords. Please provide 5-7 unique keywords that capture the essence of the study.

Introduction:
There is a spacing issue between the citation and the text in the first sentence of the introduction. Please address this to improve readability.
References: Ensure that the references used, particularly in the introduction and discussion sections, are up-to-date, ideally not older than a decade.
Abbreviation and Numbers: Avoid starting a sentence with an abbreviation or a number. If necessary, add "the" before the abbreviation or number.
Scientific Nomenclature: Italicize scientific names, such as in Line No. 123, and ensure consistency throughout the manuscript.
Abbreviations: Please include a list of abbreviations to enhance clarity and comprehension.
Materials and Methods:
Line No. 127: Provide comprehensive details about the greenhouse settings, including temperature conditions, light intensity, experimental design, replications, and soil physicochemical properties.
Line Nos. 131 and 135: Remove the connecting line in the "cohort" and "Reverse."
Line No. 132: Explain the method used to measure and maintain relative moisture content. Additionally, clarify the strategy employed for water stress imposition using PEG.
Line No. 162: Specify whether the phylogenetic tree was constructed using neighboring or non-neighboring methods, and mention that Mega 11 software was used for the analysis.
Line No. 175: Provide TPM values or include them in the supplementary material data.
Line No. 177: Provide a link or validate the authenticity of the data, or submit unpublished data to a recognized database and share the link.
Line No. 178: The phrase "How a leaf suffers stress" is unclear and should be revised for clarity.
Line No. 186: Include the figure number or diagram citation (e.g., Figure 5) to which this sentence refers.
Line No. 189: Expand the acronym "CLA gene" by providing its full form.
Line No. 205: Clarify the statement, "Before the drought treatment, cotton plants with consistent growth were selected from TRV2:00 and TRV2 lines and placed under the same conditions, being watered once every 3 days."
Line Nos. 207-208: Provide a clear description of how the stress was imposed using PEG. Specify whether the application was to the soil or foliar, and detail the quantity of PEG applied, soil moisture content, and how it was measured. Confirm that the stress was indeed water-related and not due to other factors like mineral imbalances. Also, specify the type and quantity of soil used, including its physicochemical characteristics.
Line Nos. 212-213: The sentence, "“The specific procedures for these measurements were in accordance with the instructions of the biochemical index assay kit (Grace, Suzhou, China)," requires the addition of a detailed procedure and appropriate reference.
Statistical Analysis: Provide the statistical design of the experiment, including the number of replications used. Additionally, describe the statistical analysis method at the end of the Materials and Methods section.
Line No. 217: Correct the citation style for "The lignin content was measured using a lignin content measurement kit (BC 4205; Solarbio (Liu, Luo & Zheng, 2018)."
Line No. 227: Replace "discovered" with "identified" for the CAD genes within the Gossypium hirsutum genome. Also, remove double spaces in the scientific names.
Line No. 233: The statement "while a small number were found in peroxisomes, mitochondria, and the nucleus (Table S1)" should refer to Table S2, not S1. Additionally, consider providing a heatmap of subcellular localization of genes in the main manuscript using Table S2.
Line Nos. 235-236: Justify the selection of specific plant species, such as Arabidopsis thaliana, Oryza sativa, Zea mays, and Brachypodium distachyon, in the introduction section.
Line Nos. 240-242: Confirm whether the appropriate term should be "clade" or "cohort" to avoid any potential misinterpretation.

Results and Discussion:
Line No. 303: Mention the percentage of wilting observed.
Line Nos. 303-310: Provide a technical explanation of the results and include key values that describe the percentage increase or decrease in the physiological and biochemical parameters of TRV2 compared to TRV2:00.
Line No. 334: Italicize the scientific name Glycine max and ensure that all plant scientific nomenclature throughout the manuscript is consistently italicized.
Line Nos. 336, 338, 347: Correct the terms "soy-bean" to "soybean," add a space between "Down" and "regulation," and ensure the term "pv" (pathovar) is non-italicized.
Line Nos. 348-349: The sentence "Transgenic lines overexpressing TaCAD12 exhibit stronger resistance to Sharp Eyespot Disease (Rong et al., 2016)" is irrelevant and should be removed.
Line Nos. 350 & 353: Italicize the scientific names Botrytis cinerea and Phyllostachys edulis.

Discussion: The discussion section is inadequately developed and does not fully meet the requirements. The authors should provide a more comprehensive analysis of their findings in the context of current literature. The second paragraph, which provides a general overview of the CAD gene family in other plant species, seems more appropriate for the introduction. Instead, the discussion should focus on the study's specific findings, including the phylogenetic analysis, co-expression, physiological, and biochemical parameters, as well as lignin accumulation.
Conclusion: Rewrite the conclusion section to validate the findings in broader terms, highlighting the implications and potential applications of the research.

Figures and Tables: Provide abbreviations in the footnotes of each figure to ensure they are self-explanatory. Ensure that all figures and tables are detailed enough to be understood independently.

Supplemental Information: Clearly describe the contents of the supplemental information in Line No. 396.
Future Recommendations: Consider adding a discussion or future recommendations on whether the silencing of GhiCAD genes influences other genes, as gene networks are often interconnected.
References and Citations: Double-check the accuracy and formatting of the reference list and in-text citations to ensure consistency with the journal's guidelines.

Experimental design

The information provided on the experimental design is currently insufficient to ensure reproducibility. It is strongly recommended that the authors include comprehensive details on the experimental setup, including specifics on replication, controls, environmental conditions, and statistical methods used. This will significantly enhance the reliability and reproducibility of the study's findings.

Validity of the findings

The findings seems promising.

---

## Round 0.2 · Minor Revisions

Dear Authors

The manuscript still needs a minor revision before being reconsidered for publication. The authors are invited to revise the paper considering all the suggestions made by the reviewers. Please note that the requested changes are required for publication.

With Thanks

Reviewer 1 ·

Basic reporting

I would like to thank the authors for their thorough revisions and thoughtful responses to previous comments. After reviewing the revised manuscript, I am pleased to see that all of my concerns have been adequately addressed. The authors have made the necessary changes and provided clear explanations where appropriate. I have no further comments or suggestions, and I believe the manuscript is now suitable for publication.

Experimental design

The experimental design is appropriate

Validity of the findings

The findings presented in the manuscript are valid, supported by robust data and sound statistical analysis.

Reviewer 2 ·

Basic reporting

No comment

Experimental design

No comment

Validity of the findings

No comment

Additional comments

The authors have made the changes I suggested in the last review. I recommend its publication in this journal.

·

Basic reporting

The revised manuscript addresses several queries; however, some previously raised concerns remain unresolved. Additionally, the authors are encouraged to compile the report in a more structured and organized manner, which will facilitate smoother analysis. This includes explicitly indicating the line numbers where the suggested revisions have been incorporated. Key points that require attention include:
Missing Abbreviations: The list of abbreviations is still missing.
There is no mention of the experimental design or replication details. Additionally, the authors have not provided any information regarding the soil's physico-chemical properties.
The manuscript lacks clarity on how PEG was used to induce stress. The authors must provide a detailed description of the stress imposition, specifying whether the PEG application was soil-based or foliar, the quantity of PEG applied, soil moisture content, and how this was measured. It is crucial to confirm that the stress was water-related, ruling out potential confounding factors such as mineral imbalances.
The methodology used for assessing catalase (CAT), malondialdehyde (MDA), and superoxide dismutase (SOD) activity is not provided and should be included.
The statistical methods used to validate the results are not mentioned. A detailed description of these methods is necessary for the interpretation of the data.
There are numerous typographical and grammatical errors throughout the manuscript, which need to be thoroughly reviewed and corrected.
While the method for determining relative soil water content (RSWC) is adequately explained, more detail is needed on the application of drought stress. It is unclear whether water was completely withheld or reduced incrementally. Measures ensuring uniform drought stress across the experimental pots should be described to confirm that plants experienced similar levels of stress.
The authors have not provided information on how the qPCR primers for CAD genes were designed. Clarifying whether the primers targeted specific exons or were based on existing literature would improve the clarity of this section.
The figures and supplementary tables are not extensively discussed in the results section, which limits the manuscript’s interpretative depth.

Experimental design

Clarifications are required as mentioned in basic reporting.

Validity of the findings

Validity of the findings can be decided on the provision of missing information as mentioned in basic reporting.

---

## Round 0.3 · accepted · Accept

Dear Authors,

I am pleased to inform you that after the last round of revision, the manuscript has been improved, and it can be accepted for publication.

Congratulations on the acceptance of your manuscript and thank you for your interest in submitting your work to PeerJ.

With Thanks

·

Basic reporting

I am pleased to inform that the revised version of the manuscript titled "Genome-Wide identification of the Gossypium hirsutum CAD gene family and functional study of GhiCAD23 under drought stress" has satisfactorily addressed all the revisions and comments. The authors have successfully incorporated the necessary corrections, and the manuscript now adheres to the required scientific and technical standards. I believe it is suitable for publication in PeerJ in its current form.

Experimental design

The experimental design is robust and well-structured, ensuring the reliability and accuracy of the data collected.

Validity of the findings

The findings appear promising and provide valuable insights, suggesting the potential for significant contributions to the field.